# Association of Liver Damage and Quasispecies Maturity in Chronic HCV Patients: The Fate of a Quasispecies

**DOI:** 10.3390/microorganisms12112213

**Published:** 2024-10-31

**Authors:** Josep Gregori, Marta Ibañez-Lligoña, Sergi Colomer-Castell, Carolina Campos, Damir García-Cehic, Josep Quer

**Affiliations:** 1Liver Diseases-Viral Hepatitis, Liver Unit, Vall d’Hebron Institut of Research (VHIR), Vall d’Hebron Barcelona Hospital Campus, Passeig Vall d’Hebron 119-129, 08035 Barcelona, Spain; marta.ibanez@vhir.org (M.I.-L.); sergi.colomer@vhir.org (S.C.-C.); carolina.campos@vhir.org (C.C.); 2Centro de Investigación Biomédica en Red de Enfermedades Hepáticas y Digestivas (CIBEREHD), Instituto de Salud Carlos III, Av. Monforte de Lemos, 3-5, 28029 Madrid, Spain; 3Medicine Department, Universitat Autònoma de Barcelona (UAB), Campus de la UAB, Placa Cívica, 08193 Bellaterra, Spain; 4Biochemistry and Molecular Biology Department, Universitat Autònoma de Barcelona (UAB), Campus de la UAB, Placa Cívica, 08193 Bellaterra, Spain; 5Department of Radiation Oncology, University of Cincinnati College of Medicine, Cincinnati, OH 45267, USA; facedgc@gmail.com

**Keywords:** quasispecies diversity, quasispecies maturity, quasispecies fitness, fibrosis, liver damage, antiviral treatment failure

## Abstract

Viral diversity and disease progression in chronic infections, and particularly how quasispecies structure affects antiviral treatment, remain key unresolved issues. Previous studies show that advanced liver fibrosis in long-term viral infections is linked to higher rates of antiviral treatment failures. Additionally, treatment failure is associated with high quasispecies fitness, which indicates greater viral diversity and adaptability. As a result, resistant variants may emerge, reducing retreatment effectiveness and increasing the chances of viral relapse. Additionally, using a mutagenic agent in monotherapy can accelerate virus evolution towards a flat-like quasispecies structure. This study examines 19 chronic HCV patients who failed direct-acting antiviral (DAA) treatments, using NGS to analyze quasispecies structure in relation to fibrosis as a marker of infection duration. Results show that HCV evolves towards a flat-like quasispecies structure over time, leading also to advanced liver damage (fibrosis F3 and F4/cirrhosis). Based on our findings and previous research, we propose that the flat-like fitness quasispecies structure is the final stage of any quasispecies in chronic infections unless eradicated. The longer the infection persists, the lower the chances of achieving a cure. Interestingly, this finding may also be applicable to other chronic infection and drug resistance in cancer.

## 1. Introduction

In patients with chronic hepatitis C virus (HCV) infection, the liver damage progression is associated with increased quasispecies fitness, caused by the dynamic nature of quasispecies as they explore the genetic space functional to the virus, facilitated by the low replication fidelity characteristic of RNA-dependent RNA polymerases. It has been observed with HCV in cell culture experiments that viral quasispecies diversify even in the absence of external factors driving selective pressure, with quasispecies fitness increases and lower sensitivity to antivirals, despite no previous exposure to these drugs, and without specific resistance mutations [1,2]. Similar evolutionary results were recently observed with chronically hepatitis E virus (HEV)-infected patients who failed ribavirin treatments, showing flat-like fitness quasispecies profiles [3,4].

Liver disease progression is typically assessed by measuring the level of liver fibrosis [5,6]. Liver fibrosis refers to the accumulation of scar tissue in the liver, which results from chronic liver damage and inflammation. The extent of fibrosis is a key indicator of the liver disease severity and helps guide treatment decisions and predict patient outcomes [7]. High levels of liver fibrosis, including cirrhosis and hepatocellular carcinoma (HCC), are associated with a greater likelihood of treatment failures when using direct-acting antivirals (DAAs) [7,8]. While DAA-based treatments are highly effective for HCV infections, their efficacy is reduced in patients with advanced fibrosis or cirrhosis. The extensive scarring and liver damage in these patients can hinder the effectiveness of DAAs, leading to lower rates of sustained virologic response and potential relapse after treatment [7].

Moreover, high quasispecies fitness has been associated with treatment failure [9,10,11,12], which refers to the ability of diverse viral variants within the HCV population to adapt and survive, even under the pressure of antiviral therapy. High quasispecies fitness indicates a greater genetic diversity and adaptability of the virus, which can lead to the emergence of resistant mutants that are less susceptible to DAAs, ultimately resulting in reduced treatment efficacy and higher chances of relapse. In the context of long-term HCV infection, the quasispecies fitness landscape may become relatively flat [4]. This means that several variants have similar fitness levels and can coexist without one being significantly more advantageous than others, at similar frequencies [1]. Concurrently, long-term infections result in progressive liver damage.

Quasispecies maturity refers to the evolutionary stage of a viral population characterized by high genetic diversity and adaptability. A quasispecies matures as it explores the sequence space through the production of new variants, primarily driven by the low replication fidelity of RNA-dependent RNA polymerases. The evolution of these new genomes is guided by fitness gradients, which can be internal to the quasispecies or produced by external selective pressures. As the quasispecies evolves, it progressively moves from peaked fitness landscapes, with a highly dominant haplotype, towards a flatter fitness landscape where multiple genomes with similar fitness coexist in a dynamic equilibrium. This evolution is facilitated by the vast number of synonymous substitutions and nearly neutral mutations available, which provide pathways for the quasispecies to traverse the sequence space. Fitness gradients within a quasispecies arise from the differences between the current frequencies of viral genomes and their respective fitness potentials.

The maturation process of a quasispecies is characterized by increasing genetic diversity, enhanced adaptability to various selective pressures, and a more robust and stable population structure. Maturity is an indicator of how well the quasispecies has explored the genetic space to optimize its fitness. Mature quasispecies have undergone extensive mutation and selection processes, resulting in resilient and robust populations capable of surviving and thriving in a variety of environmental conditions, including the presence of antiviral agents.

Our hypothesis in this study is that a high level of liver damage is indicative of a long-term chronic infection, which results also in flatter viral quasispecies in fitness terms. Using post-treatment quasispecies data from 19 patients who experienced failure with DAA regimens, and with fibrosis data available, we investigate the potential correlation between liver damage and quasispecies maturity and fitness. This is done by characterizing the quasispecies in these patients through indicators of quasispecies maturity [4].

## 2. Materials and Methods

### 2.1. Patients

We collected post-treatment quasispecies data from 19 patients who experienced failure with DAA regimens, with fibrosis levels F1 and above (F1: 2 patients, 6 amplicons; F2: 3 patients, 11 amplicons; F3: 5 patients, 16 amplicons; F4+: 9 patients, 34 amplicons. Genotypes and subtypes: 7 1a, 6 1b, 5 3a, 1 4d (Appendix A)). Transient elastography was measured using FibroScan®, classifying the patients by the degree of fibrosis as F1, F2, F3, and F4+ (F4+ indicates advanced cirrhosis).

The full cohort from which the 19 patients with fibrosis scores have been taken is composed of 75 patients with failed HCV treatments. The 56 patients with no fibrosis data have been partitioned into three groups according to their rank with the RLEinf evenness indicator—Reg (18), Mid (19), and Top (19)—with increasing maturity and flat-like fitness characteristics, according to their haplotype frequencies distribution.

### 2.2. Methods

#### 2.2.1. RNA Extraction, Amplification, and Deep-Sequencing

HCV-RNA was extracted and nested RT-PCR amplified using subtype-specific primers previously described [13]. Reverse transcription was performed using the OneStep RT-PCR Transcriptor Kit (Roche Applied Science, Basel, Switzerland). Nested PCR was then carried out with the KAPA HiFi HotStart PCR Kit (KAPA Biosystems, Roche, Pleasanton, CA, USA) to amplify four HCV amplicons corresponding to regions targeted by direct-acting antivirals (DAAs): one amplicon for the NS3 protease gene (444 bp), one for NS5A (387 bp), and two for NS5B (NS5B1 and NS5B2, both 396 bp each).

Amplified DNA was quantified using fluorescence with the Quant-iT Qubit dsDNA BR Assay Kit (ThermoFisher Scientific, Waltham, MA, USA). All amplicons were adjusted to equal concentrations, and the four amplicons from each individual were combined into a single tube and purified using KAPA Pure Beads magnetic beads (KAPA Biosystems, Roche, Pleasanton, CA, USA). A second fluorescence quantification was conducted with the Quant-iT Qubit dsDNA HS Assay Kit (ThermoFisher Scientific, Waltham, MA, USA), followed by another normalization step of all DNA pools to 1.5 ng/µL.

Library preparation for MiSeq sequencing followed the standardized protocol of the KAPA HyperPrep Kit (Roche Applied Science, Pleasanton, CA, USA), utilizing SeqCap Adapters A/B (Nimblegen, Roche Pleasanton, CA, USA) for patient-specific indexing. A subsequent clean-up using KAPA Pure Beads (KAPA Biosystems, Roche, Pleasanton, CA, USA) was performed to eliminate small DNA fragments that could contaminate the pools. Quality control was performed using the Agilent DNA 1000 kit and bioanalyzer (Santa Clara, CA, USA).

The pooled samples from each patient underwent final normalization to a concentration of 4 nM and were mixed into a final library pool. To accurately quantify the indexed DNA in the 4 nM library, qPCR was performed using the KAPA Library Quantification Kit (Kapa Biosystems, Roche, Pleasanton, CA, USA) according to the standard protocol, with analysis on the LightCycler480 system (Roche Applied Science, Basel, Switzerland). Prior to loading onto the MiSeq reagent cartridge (600 cycles, 2 × 300) (Illumina, San Diego, CA, USA), the final dilution and the ratio of PhiX Control V3 (Illumina, San Diego, CA, USA) to the library sample were determined.

Deep-sequencing was performed on the MiSeq platform (Illumina, San Diego, CA, USA) and data analyzed using in-house scripts [14,15]. Despite differences in length and function, the four sequenced amplicons have been, here, treated as one (see “Wilcoxon Test and Statistics” section).

#### 2.2.2. NGS Data Treatment

The fastq files from MiSeq Illumina were treated to preserve full-read integrity, completely covering the amplicon, to obtain amplicon haplotypes and corresponding frequencies. Briefly, as previously described (Colomer-Castell et al., 2023 [3]), full amplicon reads were obtained from the 2 × 300 bp paired-end reads with the help of FLASH [16], requiring a minimum overlap of 20 bp and a maximum of 10% mismatches; the reads accumulating more than 5% bp with Phred scores below Q30 were removed. The clean amplicon was finally obtained by trimming primers. Reads were collapsed into haplotypes and counts. These haplotypes and frequencies are the basis of subsequent computations. The results were computed for each strand independently, without previous abundance filtering or strand intersection [4].

#### 2.2.3. Quasispecies Maturity

Quasispecies structure data: Diversity values are computed from the amplicon haplotypes and corresponding frequencies as read counts. No minimal abundance filter is performed, so that singletons, that is haplotypes with a single read, are available, and the quasispecies may be deeply studied.

Each sample and amplicon provides two fasta files, one for the reads of each strand, forward and reverse.

To account for differences in amplicon coverage, a rarefaction process [17] is carried out, with the following criteria:Amplicons with a coverage below 18,000 are excluded from the computations.For amplicons with coverages above 18,000 and below 23,000, diversity values are computed with no rarefaction.For amplicons with coverages above 23,000, 200 cycles of subsampling with no replacement, to a reference size of 20,000 reads, are executed. Each subsampling cycle provides a size-normalized quasispecies on which each indicator is computed. The median of the 200 values of each indicator is taken as the rarefied value.The process is carried out with each fasta file, for each combination of strand, amplicon, and patient.

Finally, the rarefied values of each indicator are averaged over the two strands for each amplicon and patient.

The quasispecies maturity indicators [4] studied here: master haplotype frequency (Master), rare haplotypes load (Rare), the aggregated frequency of the top 25 haplotypes (Top25), the fraction of singletons (Singl), the relative logarithmic evenness at q = 1 (RLE1), 2 (RLE2), and infinity (RLEinf), and the evenness of the top 10 (R10) and 25 haplotypes (R25). These indicators are further described in the document Appendix A. Table 1 shows the expected level of each quasispecies maturity indicator in the two limiting cases, flat vs. regular quasispecies.

#### 2.2.4. Synonymity

In addition to characterizing quasispecies maturity through indicators based on haplotype distribution, quasispecies synonymity offers further insights into the maturation process. Synonymity emphasizes the relationship between haplotype diversity and expressed phenotypes. This feature can be quantified by calculating the frequency ratio of the master phenotype to the master haplotype and, more broadly, by comparing the frequencies of the top N phenotypes to the top N haplotypes [3,4].

#### 2.2.5. Wilcoxon Test and Adjusted *p*-Values

Statistical significance has been contrasted with the non-parametric Mann–Whitney–Wilcoxon (MWW) test [18], comparing the value of each of these indices between liver stages F2 and F1, F3 and F2, and F4+ and F3, with the alternative hypothesis, as formulated in Table 1, that higher liver damage corresponds to more mature quasispecies with flatter fitness landscapes. The set of *p*-values for each test has been adjusted with the Benjamini–Hochberg method [19].

Despite the difference in amplicon length and functionality between the four amplicons sequenced—NS3 444 bp, NS5A 387 bp, NS5B1 396 bp, and NS5B2 396bp—they are treated equally. The replication errors produced by the RNA-based RNA polymerase, NS5B, are random occurrences that may affect any position in the genome being replicated. However, in the long term, only those that are able to maintain the virus functionality will proliferate. Among the errors produced in the replication cycles, synonymous substitutions and neutral mutations will not affect, or minimally affect, functionality. This is expected to be the same in all regions of the genome. These are the paths used by the quasispecies to explore the genetic space. In our view, the differences in maturity indicators between amplicons will arise mainly because of their slightly different lengths. The objective behind not making distinctions between amplicons is having enough data points in the comparisons. We think that, on one hand, this is detrimental to obtaining significant results, because it adds extra variability, but on the other hand, that if results are significant, they will be more robust. Each amplicon is represented by the set of selected maturity indicators and flagged according to the fibrosis stage of the patient.

#### 2.2.6. Effect Size

Beyond the statistical significance provided by a *p*-value, it is important to consider the measure of effect size, which gives a more comprehensive understanding of the magnitude of the observed effect. While a *p*-value can indicate whether an effect exists, in statistical terms, it does not provide information about the size or practical importance of the observed effect.

The effect size of each indicator has been measured with the following statistics: AUC [20] values computed from the MWW test statistic, rank–biserial correlation, R_RB = 2 AUC-1 [21], and γ_0.5_, a non-parametric version of Cohen’s d statistic [22,23].

AUC values vary from 0.5 to 1, and R_RB values vary between 0 and 1, with higher values indicating higher association. The absolute value of γ_0.5_ gives the level of association, whereas its sign shows the direction of the effect; positive values correspond to higher values of the index for the condition on the left (i.e., F3 > F2), while negative values correspond to higher values of the index for the condition on the right (i.e., F3 < F2). In the document entitled Appendix A can be found definitions, equations, and an interpretation guide.

## 3. Results

### 3.1. Wilcoxon Test and Statistics

Statistical significance was contrasted with the non-parametric Mann–Whitney–Wilcoxon (MWW) test [18], comparing the value of each of these indices between liver stages F2 and F1, F3 and F2, and F4+ and F3.

Table 2 shows the metrics for the tests Stage2 vs. Stage1, resulting in adjusted *p*-values below 0.05 and AUC values above 0.65. Values of the tests for F2 and F1 are excluded as inconclusive. Benjamini–Hochberg-adjusted *p*-values of these tests are represented in logarithmic scale in Figure 1.

### 3.2. Effect Size

The effect size of quasispecies maturity indices [24,25], in the comparison of liver damage stages, was quantified by three metrics: area under the ROC curve (AUC), rank–biserial correlation (R_RB), and a robust non-parametric metric version of Cohen’s d (γ_0.5_) (Table 2, Table 3 and Table 4, and Figure 2 and Figure 3).

### 3.3. Collected Metrics of All Comparisons

Quasispecies structure indicators were sorted by decreasing AUC value, and *p*-value, AUC, R_RB, and γ_0.5_ were collected (Table 4).

In both comparisons, F3 vs. F2 and F4+ vs. F3, the sign of the effect for all statistically significant indicators matched with the expected behavior, negative for Master and Top25, positive for all other indicators. However, the effect magnitude was greater for all indicators in F3 vs. F2 than in F4+ vs. F3. In the comparison between F3 and F2, all indicators were statistically significant except for R10 and Top25R. Top25, R25, and Rare showed adjusted *p*-values below 1 × 10^−3^, and AUC values above 0.85. In the comparison between F4+ and F3, all indicators are statistically significant with adjusted *p*-values below 0.05 and AUC values above 0.65, except for Top25, Rare and Singleton. R10 and Top25R show the higher AUC values, above 0.725. Liver damage progression and associated infection time parallel the reduction in Master and Top25 frequencies, with an increase in all other indicators characteristic of more homogeneous haplotype distributions, and typical of flat-like quasispecies.

### 3.4. Boxplot and ROC Curves for Selected Indicators

As observed in the next boxplots and ROC curves of selected indices (Figure 4 and Figure 5), the comparison of F2 and F1 showed an inverted effect with respect to the comparisons F3 vs. F2 and F4+ vs. F3, which showed the expected evolution towards higher diversity and maturity in the quasispecies as liver damage progressed. The variable connecting quasispecies diversity increase and liver damage progression is infection time. It is unclear why the evolution from F1 to F2 that we observe in the quasispecies was towards limiting diversity. Our dataset included only two patients (six amplicons) with F1, which did not allow us to extract any conclusion about evolution from F1 to F2. A larger patient cohort would have helped clarify whether the observed patterns were biological or a result of random variation due to sample size.

### 3.5. Synonymity

During maturation, multiple haplotypes that expressed a limited set of functional phenotypes accumulate, resulting in high haplotype synonymity within the quasispecies. As replication errors and fitness gradients drive the exploration of genetic space primarily through synonymous haplotypes connected to functional phenotypes, more mature quasispecies typically exhibit higher levels of synonymity. Synonymity provides valuable insights into the quasispecies’ evolutionary optimization and its ability to maintain functional stability despite showing very high genetic diversity [3,4]. Figure 6 shows the relationship in frequencies between top haplotypes and top phenotypes for all amplicons.

### 3.6. Full Cohort Maturity Characteristics

The full cohort from which the 19 patients with fibrosis scores were taken is composed of 75 patients with failed HCV treatments. The 56 patients with no fibrosis data were partitioned into three groups according to their rank with the RLEinf evenness indicator—Reg (18), Mid (19), and Top (19)—with increasing maturity and flat-like fitness characteristics, according to their haplotype frequencies distribution. The full cohort of patients was characterized in an exploratory data analysis by their maturity status, ranked alongside patients with fibrosis data, using visual means to show their groupings and separability, which included boxplots, PCA scatterplots, and rank plots.

Unexpectedly, most Mid and Top patients showed quasispecies with higher maturity scores than all F4+ patients, whereas Reg patients were ranked among F2 to F4+ patients. Further information may be found in Appendix A.

## 4. Discussion

Our primary objective was to investigate whether quasispecies in chronic HCV infections evolve with fluctuating changes, spiraling around a limited diversity with different master sequences, or if they progress towards flat-like structures with unlimited diversity, characterized by several genomes at very low frequencies and no dominant haplotype, a phenomenon recently observed in two HEV patients with repeatedly failed ribavirin treatments [3,4].

The samples of the patients in this cohort correspond to samples sent to our laboratory with the purpose of obtaining recommendations for a DAA combination treatment. Any previously failed treatment may have contributed to accelerating (RBV) or decelerating (DAA) the natural evolution in the quasispecies maturation process, and thus increased the variability in our data with respect to what could be observed with all naïve patients. One of the limitations of our study is the lack of pre-treatment samples. However, it is likely that naïve patients will form a cohort with lower variability, potentially leading to a more statistically significant *p*-values. In contrast, selecting patients treated with different therapies introduces greater variability, which may make it more challenging to detect correlations. Nevertheless, when correlations are found in such a diverse cohort, they may be more robust. While including pre-treatment samples would be valuable, we believe that their absence will not significantly impact our reported results.

Moreover, mutagens contribute to acceleration of the natural evolution through the increased replication error rate, and DAAs contribute to deceleration of the evolution through inhibition. We observed this behavior with cell culture experiments ([26], and unpublished data). HCV genotypes/subtypes may follow the natural evolution at relatively different paces, although no large differences are expected, contributing also to an increase in the variability in our data. These possible extra sources of variability render our results more robust, although possibly at the expenses of increasing some *p*-values and reducing effect sizes.

The association between fibrosis score and quasispecies maturity indices is understood as a direct consequence of infection time. As liver damage progresses, so does quasispecies diversity. Nevertheless, this is a simplification, because we know that several factors contribute to quasispecies diversification and maturity, and to liver damage, beyond infection time: varying viral loads, immune pressure, bottleneck events, and previous failed therapies, among others. Despite the simplification, we expect that patients with more advanced liver damage, or longer infection times, will show more diversified and mature quasispecies, and the longer the infection time, the more akin to a flat-like quasispecies structure.

In this study, the comparison of quasispecies maturity indicators between patients with known liver damage and the rest of the cohort supports our hypothesis that as chronic infections advance, flat-like quasispecies characteristics become increasingly prevalent in the overall quasispecies structure. In our dataset, the progression of liver damage, beginning at stage F2 and onwards, shows this quasispecies maturation. Boxplots of the primary indicators illustrate this trend, while *p*-values indicate the statistical significance of the observed differences, and effect sizes quantify the magnitude of these differences in statistical terms. The difference in maturity indicator values between stages F3 and F2 is more pronounced than between stages F4+ and F3, showing a saturation effect in some indicators.

The results for cases below F2 were counterintuitive, showing an inverse behavior to what was expected, although these differences are not statistically significant. Stage F1 is the least represented fibrosis stage in our dataset. Transient elastography (FibroScan) is highly reliable for diagnosing significant fibrosis (≥F2 on the METAVIR scale), but it is less reliable for distinguishing between intermediate stages of fibrosis. Additionally, factors such as obesity, ascites, inflammation, and the operator’s experience and technique may impact FibroScan results [6].

The variability in the results increases as maturity scores become higher, as observed in characterizing the full patient cohort. This is likely caused by the longer clinical history of more advanced patients. Longer infection times allow for the emergence of different factors contributing to acceleration or deceleration of the maturation process at different time points. The accumulation of episodes caused by the factors noted above result in a higher variability in the observed maturity scores (See Appendix A).

It has been observed with in vitro studies that viral quasispecies diversify even in the absence of external factors driving selective pressure [1,2], and the reanalysis of HCV cell culture data at passages p0, p100, and p200, taken from [26], showed the same evolutionary changes as with patients with fibrosis data available. This is a simplified experiment, free of external uncontrolled factors, that further corroborates the above results (See Appendix A showing these unpublished results). These results show the evolution from a more peaked fitness landscape towards a flatter distribution, as observed using the fibrosis stage as a surrogate of infection time with clinical data. The 200 cell culture passages are equivalent to 700 days of continuous HCV replication [27]. The greatest changes have been observed between p0 and p100, with a quasispecies fitness fold-change of 2.3. The changes from p100 to p200 show the same trends, although with smaller differences, and a moderated quasispecies fitness increase.

As new mutants are generated during virus replication, due to the low-fidelity RNA-dependent RNA polymerase, they proliferate based on their replication ability, which is determined by their fitness in competition with existing genomes and the environment. Under ideal conditions, a quasispecies will continuously explore the genetic space functional to the virus, producing numerous synonymous genomes that express the same phenotype, as well as a variety of genomes with alternative but still functional phenotypes. This diversification results in a quasispecies with a progressive flat-like fitness landscape, which is the likely fate of any quasispecies. We term this evolution process quasispecies maturation. This tendency could be accelerated by the treatment with mutagenic agents, whenever the viral load remains sufficiently high [4].

Our results suggest that, during a prolonged chronic viral infection, the fate of a quasispecies is to progressively diversify, even in the absence of antiviral treatment. This leads to the generation of a highly variable population composed by a vast number of variants. The visual representation of this diversification shows that the sequence population obtained after deep-sequencing resembles a “flat population”—a structure characterized by a master sequence present at a low frequency, surrounded by a large number of different haplotypes (i.e., distinct sequences), all with similar frequencies.

This flat structure offers significant clinical advantages to the virus. It increases the robustness of the viral population, enhancing its ability to evade immune responses and making it more resilient to new antiviral treatments, particularly monotherapies, even without the presence of resistance-associated substitutions. Additionally, it facilitates escape from vaccines. In summary, this flat-structured population aids the virus in adapting to any unexpected environmental changes, promoting its survival and persistence.

A corollary to this discussion is that in HCV patients, advanced chronic infections will correspond to advanced liver damage and to flatter HCV quasispecies, associated with higher mean quasispecies fitness, and these patients will likely exhibit higher resistance to further treatments.

## 5. Conclusions

The multiple paths available to a quasispecies for exploring genetic space, in conjunction with fitness gradients and sufficient viral loads, enable continuous quasispecies evolution. Over an extended infection period, this evolution may approach a flat-like fitness landscape, characterized by several genomes existing at similar frequencies, with no dominant genome. This diverse population becomes highly resilient to future changes in the quasispecies environment, including challenges posed by direct-acting antiviral (DAA) treatments.

## Figures and Tables

**Figure 1 microorganisms-12-02213-f001:**
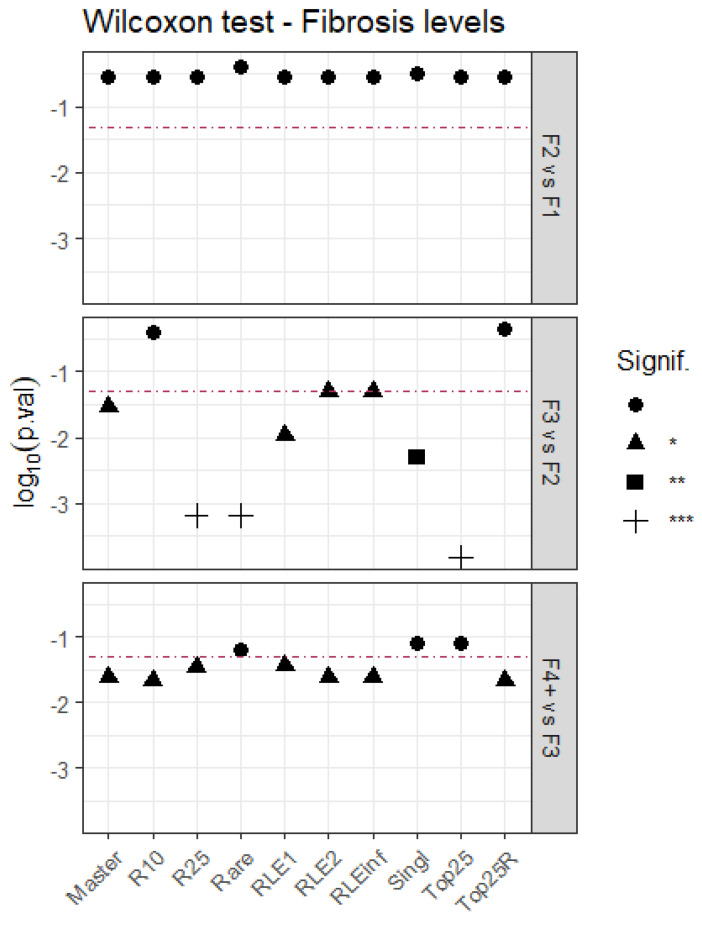
Wilcoxon test, BH-adjusted *p*-values in log10 scale. Significance: * *p*-value < 0.05, ** *p*-value < 0.01, *** *p*-value < 0.001.

**Figure 2 microorganisms-12-02213-f002:**
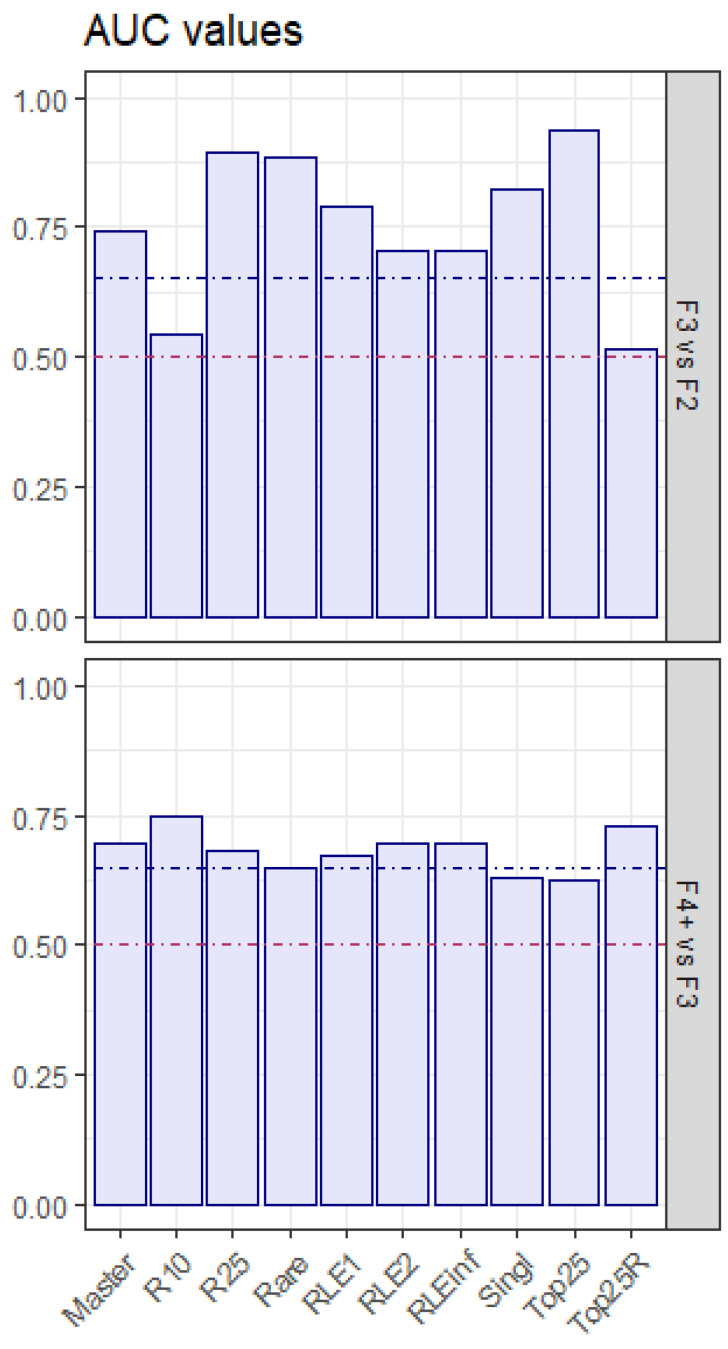
AUC values resulting from the Wilcoxon tests, with effect magnitude borders. Red line: no effect, blue line: moderate effect.

**Figure 3 microorganisms-12-02213-f003:**
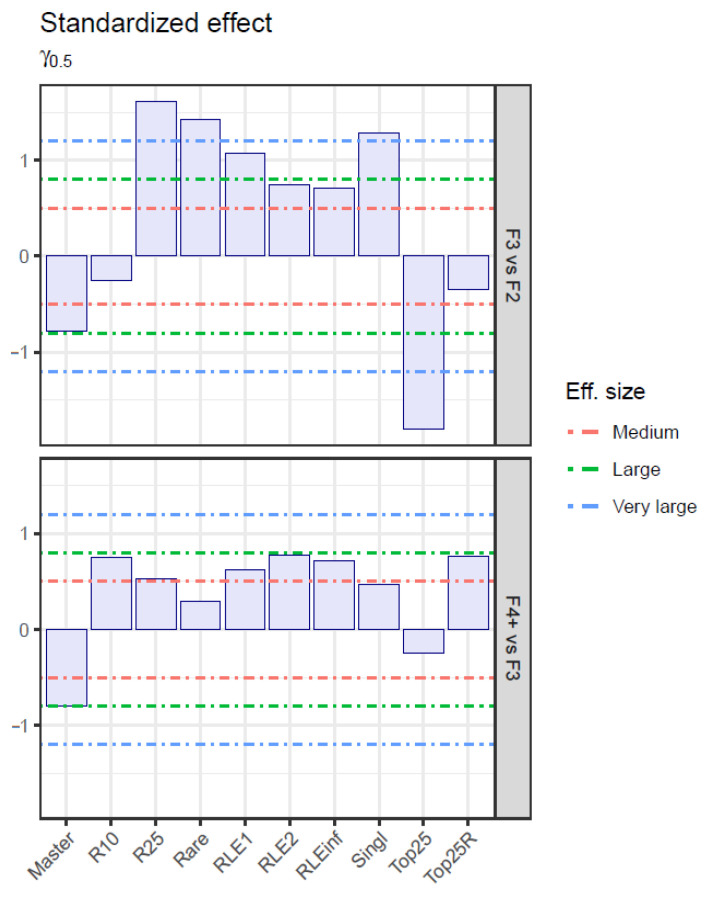
$\gamma_{0.5}$ effect size, with effect magnitude borders.

**Figure 4 microorganisms-12-02213-f004:**
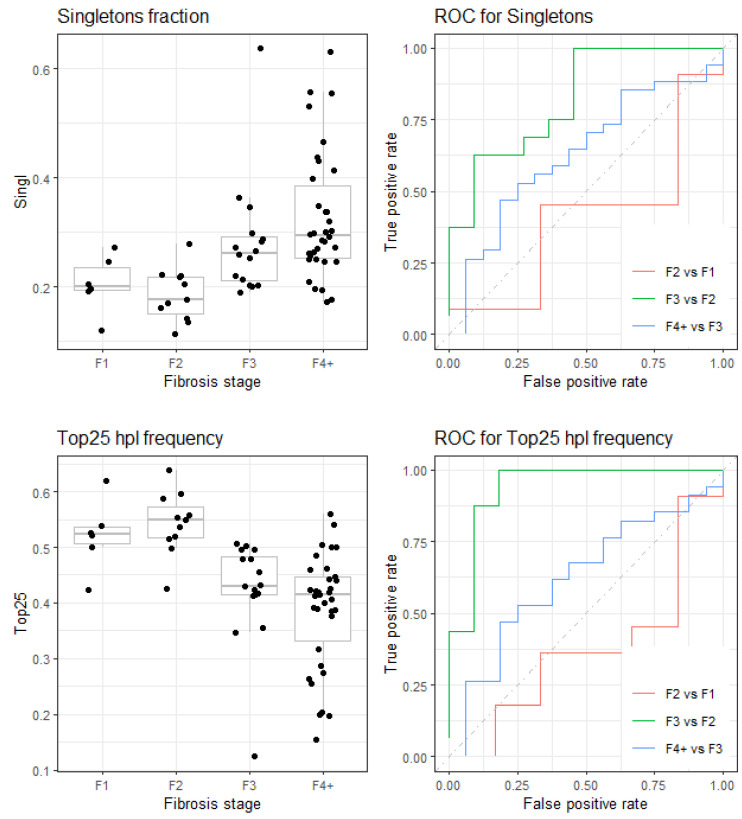
(**Top**) Fraction of singletons. (**Bottom**) Fraction of reads for top 25 haplotypes. Boxplots and ROC curves.

**Figure 5 microorganisms-12-02213-f005:**
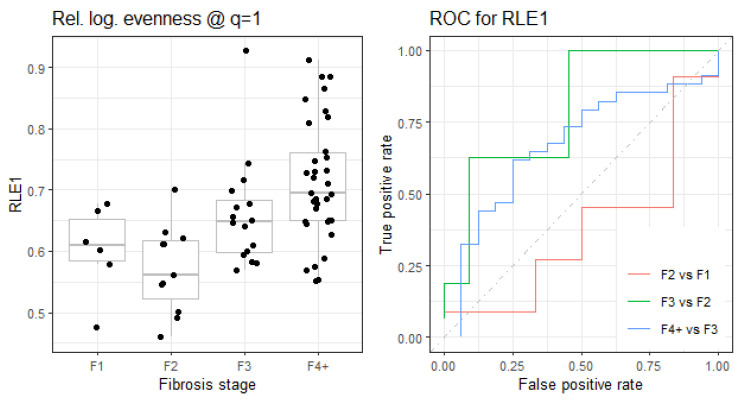
(**Top**) Relative logarithmic evenness at q = 1. (**Bottom**) Evenness among top 25 haplotypes. Boxplots and ROC curves.

**Figure 6 microorganisms-12-02213-f006:**
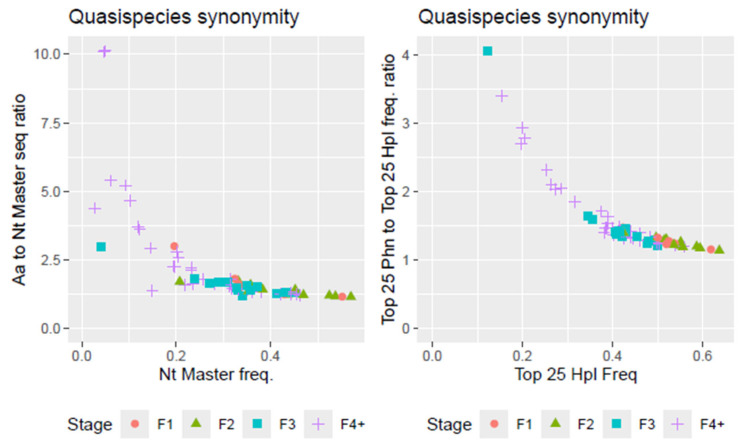
(**Left**) Quasispecies synonymity as the ratio of frequencies of top phenotype to top haplotype. (**Right**) Quasispecies synonymity as the ratio of frequencies of top 25 phenotypes to top 25 haplotypes.

**Table 1 microorganisms-12-02213-t001:** Reference table showing the expected level of each quasispecies maturity indicator in the two limiting cases, flat vs. regular quasispecies [4].

Quasispecies Feature	Description	Regular Qs	Flat Qs
Master	Frequency of the dominant haplotype	High	Low
Rare	Fraction of reads for haplotypes with frequencies <1%	Low	High
Top25	Fraction of reads for the 25 most abundant haplotypes	High	Low
Top25R	Ratio of frequencies Top25 to Master	Low	High
Singletons	Fraction of reads for singletons, haplotypes with a single read	Low	High
RLE1	Relative logarithmic evenness at *q* = 1	Low	High
RLE2	Relative logarithmic evenness at *q* = 2	Low	High
RLEinf	Relative logarithmic evenness at *q* = ∞	Low	High
R10	Evenness among top 10 haplotypes	Low	High
R25	Evenness among top 25 haplotypes	Low	High

**Table 2 microorganisms-12-02213-t002:** Wilcoxon tests results, Stage2 vs. Stage1. Effect size as the area under the ROC curve (AUC), *p*-values, Benjamini–Hochberg-adjusted *p*-values, and statistical significance. Significance: * *p*-value < 0.05, ** *p*-value < 0.01, *** *p*-value < 0.001.

Stage2	Stage1	Quasispecies Feature	AUC	*p*-Value	BH-Adjusted	Significance
F3	F2	Master	0.7443	1.715 × 10^−2^	2.858 × 10^−2^	*
F3	F2	Rare	0.8864	1.927 × 10^−4^	6.425 × 10^−4^	***
F3	F2	Top25	0.9375	1.496 × 10^−5^	1.496 × 10^−4^	***
F3	F2	Singletons	0.8239	1.961 × 10^−3^	4.903 × 10^−3^	**
F3	F2	RLE1	0.7898	5.429 × 10^−3^	1.086 × 10^−2^	*
F3	F2	R25	0.8920	1.503 × 10^−4^	6.425 × 10^−4^	***
F4+	F3	Master	0.6985	1.213 × 10^−2^	2.425 × 10^−2^	*
F4+	F3	Top25R	0.7298	4.299 × 10^−3^	2.150 × 10^−2^	*
F4+	F3	RLE2	0.6985	1.213 × 10^−2^	2.425 × 10^−2^	*
F4+	F3	RLEinf	0.6985	1.213 × 10^−2^	2.425 × 10^−2^	*
F4+	F3	R10	0.7482	2.171 × 10^−3^	2.150 × 10^−2^	*
F4+	F3	R25	0.6801	2.086 × 10^−2^	3.476 × 10^−2^	*

**Table 3 microorganisms-12-02213-t003:** Robust Cohen’s effect size $\gamma_{0.5}$.

Quasispecies Feature	F3 vs. F2	F4+ vs. F3
Master	−0.7837	−0.7998
Rare	1.4244	0.2939
Top25	−1.8065	−0.2523
Top25R	−0.3433	0.7599
Singletons	1.2839	0.4623
RLE1	1.0706	0.6176
RLE2	0.7429	0.7685
RLEinf	0.7132	0.7161
R10	−0.2475	0.7519
R25	1.6165	0.5243

**Table 4 microorganisms-12-02213-t004:** Comparison results, adjusted *p*-values, and effect size metrics. Sorted by decreasing AUC values, with (a) comparing F3 and F2, and (b) comparing F4+ and F3. Positive γ_0.5_ corresponds to greater values of the indicator for the left class (i.e., F3 > F2), negative γ_0.5_ corresponds to greater values of the indicator for the right class (i.e., F3 < F2). Significance: * *p*-value < 0.05, ** *p*-value < 0.01, *** *p*-value < 0.001.

(a) F3 vs. F2					
Quasispecies Feature	Adjusted *p*-Value	Significance	AUC	R_RB_	γ_0.5_
Top25	1.50 × 10^−4^	***	0.938	0.875	−1.810
R25	6.43 × 10^−4^	***	0.892	0.784	1.620
Rare	6.43 × 10^−4^	***	0.886	0.773	1.420
Singleton	4.90 × 10^−3^	**	0.824	0.648	1.280
RLE1	1.09 × 10^−2^	*	0.790	0.580	1.070
Master	2.86 × 10^−2^	*	0.744	0.489	−0.784
RLE2	4.98 × 10^−2^	*	0.704	0.409	0.743
RLEinf	4.98 × 10^−2^	*	0.704	0.409	0.713
R10	3.98 × 10^−1^		0.546	0.091	−0.247
Top25R	4.52 × 10^−1^		0.517	0.034	−0.343
**(b) F4+ vs. F3**					
**Quasispecies Feature**	**Adjusted** ***p*-Value**	**Significance**	**AUC**	**R_RB_**	**γ** ** _0.5_ **
R10	2.15 × 10^−2^	*	0.748	0.496	0.752
Top25R	2.15 × 10^−2^	*	0.730	0.460	0.760
Master	2.43 × 10^−2^	*	0.698	0.397	−0.800
RLE2	2.43 × 10^−2^	*	0.698	0.397	0.768
RLEinf	2.43 × 10^−2^	*	0.698	0.397	0.716
R25	3.48 × 10^−2^	*	0.680	0.360	0.524
RLE1	3.65 × 10^−2^	*	0.673	0.346	0.618
Rare	6.16 × 10^−2^		0.647	0.294	0.294
Singleton	7.99 × 10^−2^		0.630	0.261	0.462
Top25	8.10 × 10^−2^		0.625	0.250	−0.252

## Data Availability

Data are contained within the article and Appendix A. The data that support the findings of this study are openly available in the GenBank Sequence Read Archive database with BioProject accession PRJNA1176499.

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
