# Peer review of "Association of Liver Damage and Quasispecies Maturity in Chronic HCV Patients: The Fate of a Quasispecies"

_microorganisms, 2024, doi:10.3390/microorganisms12112213_

Round 1

Reviewer 1 Report

Comments and Suggestions for Authors

In this interesting manuscript the authors use HCV NS3, NS5A and NS5B deep-sequencing NGS data to determine the viral quasispecies maturity in post-treatment samples from 19 infected patients who failed antiviral therapy with DAA regimes. The topic is of general interest to improve future antiviral strategies, the methods are sounding, and the results in line with the main hypothesis. The manuscript however is not an easy read, and some important points need to be addressed.

MAJOR POINTS

The use of only post-treatment samples is the main drawback of this study, because you are no able to distinguish if changes in the quasispecies were produced by the treatment itself and not by the liver fibrosis stage of the patients. How was the quasispecies maturity in pre-treatment samples? Did you Deep-sequenced those samples? Did the authors analyze HCV quasispecies maturity in pre-treatment samples from patients with DAA response?

Another drawback is the heterogeneity of viral genotypes in the study cohort (7 gt-1a; 6 gt-1b; 5 gt-3a; 1 gt-4d), because the behavior of the quasispecies may vary for some HCV genotypes. This important limitation deserves, at least, a comment in the discussion. Did the quasispecies maturity parameters differ between HCV genotpes/subtytpes?

The methodology section is confusing, especially the wet lab part, detalis on the lab methods are lacking in the methods section, but later explained in the results: PCR amplification, amplicon length (shown in L165-169), pooling strategy, library prep (nextera?).

L142. I believe some brief description of the quasispecies indicators (at least the variable names) may be provided here in benefit of the reader.

L169. You need to further justify why the analysis is treating all gene segments as the same even though they have different evolutionary pressures.

In paragraph 3.3 (L193) add a paragraph describing the main findings on Effect size.

In paragraph 3.4 (L216) add a paragraph describing the main findings on “all the comparisons”.

Paragraph 3.6 should be excluded from the main body, and can be added to the discussion section (along L302-307) as unpublished data correlating with the results found here. Otherwise, it is very confusing finding these cell culture-derived data results when the study did not include cell culture.

L251-255. This should be moved to methods section, to clearly define the initial patient cohort, the selected patients for this study, and the partition/classification.. the classification description here is confusing.

L264-269 and 279-288. Here, the authors need to discuss the limitations on 1) not using Deep sequencing data from pre-treatment samples; 2) analyzing diffrent HCV subtypes.

MINOR POINTS

L160. Table 1 headings, I recommend you change “Indicator” for “Quasispecies feature” and not using abbreviations in the column (such as “Singl”)

L188. Table 2 headings, I recommend not using abbreviations in the headings (Stg2, 1, Feat…etc.) and in the Quasispecies Feature column. For “Star” I guess you mean significance level? – same applies to Fig.1

L208. Table 3 headings: the caption is not self-explanatory. Change “Feat” for “Quasispecies feature”

L208. Table 4 headings: the caption is not self-explanatory. Change “Test” for “Comparison” “Feat” for “Quasispecies feature”, “Signif.” For “Significance”. Do not abbreviate variable names in column “Quasispecies feature”.

L105. Are these scripts available in GitHub?

L222. …inverted effect?

L227…more patient data could be required?

L297. ..longer chronic infection timeframe on patients with advanced fibrosis?

L310. …with existing genomes and the environment?

Comments on the Quality of English Language

L145, 162, 174, etc. I recommend you use past simple tense when describing the results.

Check spelling in lines 56, 62, 121, 128.

Author Response

In this interesting manuscript the authors use HCV NS3, NS5A and NS5B deep-sequencing NGS data to determine the viral quasispecies maturity in post-treatment samples from 19 infected patients who failed antiviral therapy with DAA regimes. The topic is of general interest to improve future antiviral strategies, the methods are sounding, and the results in line with the main hypothesis. The manuscript however is not an easy read, and some important points need to be addressed.

MAJOR POINTS

The use of only post-treatment samples is the main drawback of this study, because you are no able to distinguish if changes in the quasispecies were produced by the treatment itself and not by the liver fibrosis stage of the patients. How was the quasispecies maturity in pre-treatment samples? Did you Deep-sequenced those samples? Did the authors analyze HCV quasispecies maturity in pre-treatment samples from patients with DAA response?

Answer:  We agree with the reviewer that these points should be clarified. We have associated maturity indicators with fibrosis scores, using liver damage as a surrogate of infection time, with the ultimate goal of linking maturity indicators to infection time. In this study, we report that the longer the infection persists, the more mature the quasispecies become. 

To clarify this further, we have included a paragraph at the beginning of Discussion section (lines 353 to 361 of the clean version), emphasizing this idea:

“The association between fibrosis score and quasispecies maturity indices is understood as a direct consequence of infection time. As liver damage progresses, so does quasispecies diversity. Nevertheless, this is a simplification, because we know that several factors contribute to quasispecies diversification and maturity, and to liver damage, beyond infection time: varying viral loads, immune pressure, bottleneck events, and previous failed therapies, among others. Despite the simplification, we expect that patients with more advanced liver damage, or longer infection times, will show more diversified and mature quasispecies, and as longer the infection time, the more akin to a flat-like quasispecies structure.”

In addition, we have included two paragraphs addressing the reviewer’s comments regarding the lack of naïve pre-treatment samples. Please refer to lines 333 to 352 of the clean version:

“The samples of the patients in this cohort correspond to samples sent to our laboratory with the purpose to get recommendations for a DAA combination treatment. Any previously failed treatment may have contributed to accelerate (RBV) or decelerate (DAA) the natural evolution in the quasispecies maturation process, and thus contribute to increase the variability in our data with respect to what could be observed with all naïve patients. One of the limitations of our study, is the lack of pre-treatment samples. However, it is likely that naive patients will form a cohort with lower variability, potentially leading to a more statistically significant p-values. In contrast, selecting patients treated with different therapies introduces greater variability, which may make it more challenging to detect correlations. Nevertheless, when correlations are found in such a diverse cohort, they may be more robust. While including pre-treatment samples would be valuable, we believe their absence will not significantly impact our reported results.

Moreover, mutagens contribute to accelerate the natural evolution through the in-creased replication error rate, DAAs contribute to decelerate the evolution through inhibition. We observed this behavior with cell culture experiments ([26], and unpublished data). HCV genotypes/subtypes may follow the natural evolution at relatively different paces, although not big differences are expected, contributing also to increase the variability in our data. These possible extra sources of variability render our results more robust, although possibly at the expenses of increasing some p-values and reducing effect sizes.”

Additionally, we have included a table in the main supplementary materials containing the available clinical data of the 19 patients used in the association study (Table S1).

Another drawback is the heterogeneity of viral genotypes in the study cohort (7 gt-1a; 6 gt-1b; 5 gt-3a; 1 gt-4d), because the behavior of the quasispecies may vary for some HCV genotypes. This important limitation deserves, at least, a comment in the discussion. Did the quasispecies maturity parameters differ between HCV genotpes/subtytpes?

Answer: Thank you for the pertinent comment. The evolution of a quasispecies towards higher fitness, unless eradicated, is generally expected to follow a similar patterns across all subtypes. However, the rate at which this evolution occurs may differ between subtypes, which could contribute to increased variability in the data and reduce the power of the comparisons. We have added an explanation in the discussion section lines 348 to 352 in the clean version:  “HCV genotypes/subtypes may follow the natural evolution at relatively different paces, although not big differences are expected, contributing also to increase the variability in our data. These possible extra sources of variability render our results more robust, alt-hough possibly at the expenses of increasing some p-values and reducing effect sizes”.

The methodology section is confusing, especially the wet lab part, details on the lab methods are lacking in the methods section, but later explained in the results: PCR amplification, amplicon length (shown in L165-169), pooling strategy, library prep (nextera?).

Answer:  Thank you very much for the comment. We have included detailed descriptions of our laboratory methods to enhance clarity. See lines 112 to 140 in the clean version: “Reverse transcription was performed using the OneStep RT-PCR Transcriptor Kit (Roche Applied Science, Basel, Switzerland). Nested PCR was then carried out with the KAPA HiFi HotStart PCR Kit (KAPA Biosystems, Roche, Pleasanton, CA, USA) to amplify four HCV amplicons corresponding to regions targeted by direct-acting antivirals (DAAs): one amplicon for the NS3 protease gene (444 bp), one for NS5A (387 bp), and two for NS5B (NS5B1 and NS5B2, both 396 bp each).

Amplified DNA was quantified using fluorescence with the Quant-iT Qubit dsDNA BR Assay Kit (ThermoFisher Scientific, Waltham, Massachusetts, USA). All amplicons were adjusted to equal concentrations, and the four amplicons from each individual were combined into a single tube and purified using KAPA Pure Beads magnetic beads (KAPA Biosystems, Roche, Pleasanton, CA, USA). A second fluorescence quantification was conducted with the Quant-iT Qubit dsDNA HS Assay Kit (ThermoFisher Scientific, Waltham, Massachusetts, USA), followed by another normalization step of all DNA pools to 1.5 ng/µL.

Library preparation for MiSeq sequencing followed the standardized protocol of the KAPA HyperPrep Kit (Roche Applied Science, Pleasanton, CA, USA), utilizing SeqCap Adapters A/B (Nimblegen, Roche Pleasanton, CA, USA) for patient-specific indexing. A subsequent clean-up using KAPA Pure Beads (KAPA Biosystems, Roche, Pleasanton, CA, USA) was performed to eliminate small DNA fragments that could contaminate the pools. Quality control was performed using the Agilent DNA 1000 kit and bioanalyzer (Santa Clara, CA, USA).

The pooled samples from each patient underwent final normalization to a concentration of 4 nM, and were mixed into a final library pool. To accurately quantify the indexed DNA in the 4 nM library, qPCR was performed using the KAPA Library Quantification Kit (Kapa Biosystems, Roche, Pleasanton, CA, USA) according to the standard protocol, with analysis on the LightCycler480 system (Roche Applied Science, Basel, Switzerland). Prior to loading onto the MiSeq reagent cartridge (600 cycles, 2x300) (Illumina, San Diego, CA, USA), the final dilution and the ratio of PhiX Control V3 (Illumina, San Diego, CA, USA) to the library sample were determined.”

L142. I believe some brief description of the quasispecies indicators (at least the variable names) may be provided here in benefit of the reader.

Answer:  Thank you very much for the comment. Please refer to lines 176-182 in the clean version, for a further explanation on the terms commented: “The quasispecies maturity indicators [4] studied here: master haplotype frequency (Master), rare haplotypes load (Rare), the aggregated frequency of the top 25 haplotypes (Top25), the fraction of singletons (Singl), the relative logarithmic evenness at q=1 (RLE1), 2 (RLE2) and infinity (RLEinf), and the evenness of the top 10 (R10) and 25 haplotype (R25). These indicators are further described in the document “Main Supplementary Materials.pdf”. Table 1 shows the expected level of each quasispecies maturity indicator in the two limiting cases, flat vs regular quasispecies.”

L169. You need to further justify why the analysis is treating all gene segments as the same even though they have different evolutionary pressures.

Answer:  Thank you for the comment. To address this comment, we have added a paragraph in lines 200-213 in the clean version: “The replication errors produced by the RNA-based RNA polymerase, NS5B, are random occurrences that may affect any position in the genome being replicated. However, in the long term, only those than are able to maintain the virus functionality will proliferate. Among the errors produced in the replication cycles, synonymous substitutions and neutral mutations will not affect, or minimally affect functionality. And this is expected to be same in all regions of the genome. These are the paths used by the quasispecies to explore the genetic space. In our view the differences in maturity indicators between amplicons will arise mainly because of their slightly different length. The objective behind not making distinctions between amplicons is having enough data points in the comparisons. We think that, on one hand this is detrimental to obtain significant results, because adds extra variability, but on the other hand that if results are significant they will be more robust. Each amplicon is represented by the set of selected maturity indicators, and flagged according to the fibrosis stage of the patient.”

In addition, to further support the synonymity hypothesis, a subsection with title 2.2.4 Synonymity has been added in Methods section (lines 185-191 in the clean version) and in Results section, see subsection 3.5 Synonymity (lines 302-309 in the clean version).

In paragraph 3.3 (L193) add a paragraph describing the main findings on Effect size.

Answer:  Thank you for the comment. Interpretation of Effect size is included in the Supplementary material, an indicated in lines 228-230 in the clean version: “See the document entitled Main Supplementary Materials.pdf with definitions, equations, and interpretation guide.”

In paragraph 3.4 (L216) add a paragraph describing the main findings on “all the comparisons”.

Answer:  Done. Please refer to lines 272-282 in the clean version: “In both comparisons, F3 vs F2 and F4+ vs F3, the sign of the effect for all statistically significant indicators matched with the expected behavior, negative for Master and Top25, positive for all other indicators. However, the effect magnitude was greater for all indicators in F3 vs F2 than in F4+ vs F3.  In the comparison F3 vs F2, all indicators were statistically significant except for R10 and Top25R. Top25, R25 and Rare showed adjusted p-values below 1.e-3, and AUC values above 0.85. In the comparison F4+ vs F3, all indicators are statistically significant with adjusted p-values below 0.05 and AUC values above 0.65, except for Top25, Rare and Singleton, R10 and Top25R show the higher AUC values, above 0.725. Liver damage progression, and associated infection time, paralleled the reduction in Master and Top25 frequencies, with an increase in all other indicators’ characteristic of more homogeneous haplotype distributions, and typical of flat-like quasispecies.”

Paragraph 3.6 should be excluded from the main body, and can be added to the discussion section (along L302-307) as unpublished data correlating with the results found here. Otherwise, it is very confusing finding these cell culture-derived data results when the study did not include cell culture.

Answer:  Thank you for the suggestion. The paragraph has been moved to lines 391-399 in the clean version. To clarify the reading, the unpublished results have been included in the supplementary document Supplementary material cell culture data.pdf,  as follows: “See the supplementary document “Supplementary material cell culture data.pdf” showing these unpublished results. These results show the evolution from a more peaked fitness landscape towards a flatter distribution, as observed using the fibrosis stage as surrogate of infection time with clinical data. The 200 cell culture passages are equivalent to 700 days of continuous HCV replication [27]. The biggest changes have been observed between p0 and p100, with a quasispecies fitness fold-change of 2.3. The changes from p100 to p200 show the same trends, although with smaller differences, and a moderated quasispecies fitness increase.”

L251-255. This should be moved to methods section, to clearly define the initial patient cohort, the selected patients for this study, and the partition/classification.. the classification description here is confusing.

Answer:  We agree with reviewer, and the text has been moved to methods section, lines 104-108 on the clean version:  “Section 2.1 Patients: The full cohort from where the 19 patients with fibrosis scores have been taken, is composed of 75 patients, with failed HCV treatments. The 56 patients with no fibrosis data have been partitioned in three groups according to their rank with the RLEinf evenness indicator. Reg (18), Mid (19), and Top (19), with increasing maturity and flat-like fitness characteristics, according to their haplotype frequencies distribution.”

L264-269 and 279-288. Here, the authors need to discuss the limitations on 1) not using Deep sequencing data from pre-treatment samples; 2) analyzing different HCV subtypes.

Answer:  Thank you for the comment. The following text including the main limitation of the study has been incorporated in the discussion section, lines 333-352 in the clean version: “The samples of the patients in this cohort correspond to samples sent to our laboratory with the purpose to get recommendations for a DAA combination treatment. Any previously failed treatment may have contributed to accelerate (RBV) or decelerate (DAA) the natural evolution in the quasispecies maturation process, and thus contribute to increase the variability in our data with respect to what could be observed with all naïve patients. One of the limitations of our study, is the lack of pre-treatment samples. However, it is likely that naive patients will form a cohort with lower variability, potentially leading to a more statistically significant p-values. In contrast, selecting patients treated with different therapies introduces greater variability, which may make it more challenging to detect correlations. Nevertheless, when correlations are found in such a diverse cohort, they may be more robust. While including pre-treatment samples would be valuable, we believe their absence will not significantly impact our reported results.

Moreover, mutagens contribute to accelerate the natural evolution through the in-creased replication error rate, DAAs contribute to decelerate the evolution through inhibition. We observed this behavior with cell culture experiments (26, and unpublished data). HCV genotypes/subtypes may follow the natural evolution at relatively different paces, although not big differences are expected, contributing also to increase the variability in our data. These possible extra sources of variability render our results more robust, although possibly at the expenses of increasing some p-values and reducing effect sizes.”

MINOR POINTS

L160. Table 1 headings, I recommend you change “Indicator” for “Quasispecies feature” and not using abbreviations in the column (such as “Singl”)

Answer:  Done. See Table 1, lines 183-184 in the clean version

L188. Table 2 headings, I recommend not using abbreviations in the headings (Stg2, 1, Feat…etc.) and in the Quasispecies Feature column. For “Star” I guess you mean significance level? – same applies to Fig.1

Answer:  Done. See Table 2, line 242 in the clean version

L208. Table 3 headings: the caption is not self-explanatory. Change “Feat” for “Quasispecies feature”

Answer:  Done. See Table 3, line 255 in the clean version

L208. Table 4 headings: the caption is not self-explanatory. Change “Test” for “Comparison” “Feat” for “Quasispecies feature”, “Signif.” For “Significance”. Do not abbreviate variable names in column “Quasispecies feature”.

Answer:  Done. See Table 4a and 4b, lines 263-271 in the clean version

L105. Are these scripts available in GitHub?

Answer:  No, but we are open to provide the scripts to any researcher showing interest in using them. It consists in a few thousands of R code, still lacking manual and full documentation, which complicates their use by third parties.

L222. …inverted effect?

Answer:  Done. Please refer to lines 285 in the clean version

L227…more patient data could be required?

Answer:  We have extended the explanation on whether we cannot extract any conclusion about the evolution from F1 to F2 due to the low number of patients with F1 included in our dataset. Please refer to lines 290 to 293 in the clean version: “Our data set includes only two patients (6 amplicons) with F1 which do not allow to extract any conclusion about evolution from F1 to F2. A larger patient cohort would help clarify whether the observed patterns are biological or a result of random variation due to sample size.”

L310. …with existing genomes and the environment?

Answer:  Done. Please refer to line 402 in the clean version

Comments on the Quality of English Language

L145, 162, 174, etc. I recommend you use past simple tense when describing the results.

Answer:  Done

See line 233 in the clean version:“was contrasted”

See line 248 in the clean version: “was quantified”

See lines 260-261 in the clean version: ”Quasispecies structure indicators were sorted by decreasing AUC value, and p-value, AUC, R_{BR} and γ0.5 were collected”

See line 273 in the clean version: “matched”

See line 274 in the clean version: “was greater”

See line 275 in the clean version: “were”

See line 280 in the clean version: “paralleled”

See line 285 in the clean version: “showed”

See line 286 in the clean version: “showed”

See line 287 in the clean version: “progressed”

See line 289 in the clean version: “was towards”

See line 290 in the clean version: “included”

See line 290 in the clean version: “did not”

See line 292 in the clean version: “would have helped clarify whether the observed patterns were biological or a result of”

See line 303 in the clean version: “expressed”

See line 305 in the clean version: “drove”

See line 307in the clean version: “provided”

See line 314 in the clean version: “were taken”

See line 316 in the clean version: “were partitioned”

See line 319 in the clean version: “was characterized”

See line 322 in the clean version: “showed”

See line 323 in the clean version: “were ranked”

Check spelling in lines 56, 62, 121, 128.

Answer:  Done

scar-ring was corrected by scarring in line 56 in the clean version

adapt-ability was corrected by adaptability in line 62 in the clean version

inter-section was corrected by intersection in line 155 in the clean version

re-verse was corrected by reverse in line 162 in the clean version

Reviewer 2 Report

Comments and Suggestions for Authors

The authors present a novel and very interesting manuscript. The introduction section is well written providing enough data to the reader, in order to follow the rest of the paper. However, it should be clearly noted that in the DAA era treatment failure is highly uncommon in all patients.

The materials and methods section and the results section, seem quite mixed up. The first 2-3 paragraphs of the results section look like they belong to materials and methods one. Both section should be re-written and the results section should be mopre straightforward. Explaining the statistical procedure should be in the methods section, not in each paragraph in the result one. By doing that, the reader will more easily comprehend the manuscript. 

Lastly, the discussion section should try to give pathophysiological explanations for the authors' findings, insted of re-showing the results. Lastly, a limitations part should be added

Comments on the Quality of English Language

English quality fine

Author Response

The authors present a novel and very interesting manuscript. The introduction section is well written providing enough data to the reader, in order to follow the rest of the paper. However, it should be clearly noted that in the DAA era treatment failure is highly uncommon in all patients.

The materials and methods section and the results section, seem quite mixed up. The first 2-3 paragraphs of the results section look like they belong to materials and methods one. Both section should be rewritten and the results section should be more straightforward. Explaining the statistical procedure should be in the methods section, not in each paragraph in the result one. By doing that, the reader will more easily comprehend the manuscript. 

Answer:  We would like to thank reviewer for the comments. Following reviewer’s indications we have make several changes, to improve readability. We have significantly improved material and methods section and results and make the proposed rearrangements. Previous section 3.1 has been moved to section 2.2.3 in Methods (lines 176 to 182 in the clean version); previous lines 165 to 172 in section 3.2, have been moved to section 2.2.5 in Methods (lines 199 to 213 in the clean version); previous lines 178 to 182 in section 3.2 have been moved to section 2.2.6 (lines 220 to 229 in the clean version); table 2 has been moved from section 3.2 to 3.1 (lines 240 to 242 in the clean version); and previous lines 194 to 197 in section 3.3, have been moved to section 2.2.6. (lines 215 to 219 in the clean version)

We are sure that with the changes proposed by the reviewer the manuscript has significantly improved.

Lastly, the discussion section should try to give pathophysiological explanations for the authors' findings, instead of re-showing the results.

Answer:  Thank you very much for the pertinent comment. We have added into discussion section a full paragraph giving pathophysiological explanations in lines 410-422 in the clean version: 

“Our results suggest that, during a prolonged chronic viral infection, the fate of a quasispecies is to progressively diversify, even in the absence of antiviral treatment. This leads to the generation of a highly variable population composed by a vast number of variants. The visual representation of this diversification shows that the sequence population obtained after deep-sequencing resembles a “flat population” -a structure characterized by a master sequenced present at a low frequency, surrounded by a large number of different haplotypes (i.e., distinct sequences), all with similar frequencies.

This flat structure offers significant clinical advantages to the virus. It increases the robustness of the viral population, enhancing its ability to evade immune responses and making it more resilient to new antiviral treatments, particularly monotherapies, even without the presence of resistance-associate substitutions. Additionally, it facilitates escape from vaccines. In summary, this flat-structured population aids the virus adapting to any unexpected environmental changes, promoting its survival and persistence.”

Lastly, a limitations part should be added

Answer:  We agree with reviewer that limitations should be clearly stated in the manuscript. To correct this gap, we have added a full paragraph in lines 333 to 352 in the clean version discussing limitations and factors affecting the variability in the results.

“The samples of the patients in this cohort correspond to samples sent to our laboratory with the purpose to get recommendations for a DAA combination treatment. Any previously failed treatment may have contributed to accelerate (RBV) or decelerate (DAA) the natural evolution in the quasispecies maturation process, and thus contribute to increase the variability in our data with respect to what could be observed with all naïve patients. One of the limitations of our study, is the lack of pre-treatment samples. However, it is likely that naive patients will form a cohort with lower variability, potentially leading to a more statistically significant p-values. In contrast, selecting patients treated with different therapies introduces greater variability, which may make it more challenging to detect correlations. Nevertheless, when correlations are found in such a diverse cohort, they may be more robust. While including pre-treatment samples would be valuable, we believe their absence will not significantly impact our reported results.

Moreover, mutagens contribute to accelerate the natural evolution through the in-creased replication error rate, DAAs contribute to decelerate the evolution through inhibition. We observed this behavior with cell culture experiments (26, and unpublished data). HCV genotypes/subtypes may follow the natural evolution at relatively different paces, although not big differences are expected, contributing also to increase the variability in our data. These possible extra sources of variability render our results more robust, although possibly at the expenses of increasing some p-values and reducing effect sizes.”

Reviewer 3 Report

Comments and Suggestions for Authors

I have read with interest the manuscript "Association of liver damage and quasispecies maturity in chronic HCV patients: The fate of a quasispecies" by Gregoi and collaborators. This study aimed to prove that a high level of liver damage indicates long-term chronic infection, resulting in flatter viral quasispecies in fitness terms. The paper is generally well-organized and accurately reflects the study. There are some comments and recommendations that may improve the paper.

In the Introduction, it is reasonable to have the aim of the study at the end of this section; here, the hypothesis is presented in the middle of the section, and then the definition and consideration regarding the quasispecies maturity are presented.

The authors must improve the patients' presentations and the way they evaluate fibrosis. What stages are F4+? Are there more stages than cirrhosis (F4)? There are no data regarding the sex, age, or other lab results of the patients.

Some data presented in the Results section are more appropriate to be in the Methodology section. 

Table 2 may be changed. Instead of stg1 and stg 2, this may be split into 2 tables, and the comparison F3/F2 and F4/F3 results may be presented in each table.

All tables must include the definition of the abbreviated words.

Regarding the figures with AUC values, it would probably be significant to draw the level that shows the relevance of the AUC value for diagnostic purposes. 

The authors should revise the text in lines 213 and 214 as there is no sentence there.

It would be recommended not to use Fobroscan (as in line 291) but transient elastography.

The authors must include a paragraph describing their study's strengths and limitations. They may also discuss how the results will influence the management and care of these patients. 

Comments on the Quality of English Language

I did not find significant errors regarding the language or editing.

Author Response

I have read with interest the manuscript "Association of liver damage and quasispecies maturity in chronic HCV patients: The fate of a quasispecies" by Gregoi and collaborators. This study aimed to prove that a high level of liver damage indicates long-term chronic infection, resulting in flatter viral quasispecies in fitness terms. The paper is generally well-organized and accurately reflects the study. There are some comments and recommendations that may improve the paper.

In the Introduction, it is reasonable to have the aim of the study at the end of this section; here, the hypothesis is presented in the middle of the section, and then the definition and consideration regarding the quasispecies maturity are presented.

Answer:  The paragraph with the hypothesis has been moved to the end of the introduction, see lines 90 to 95 in the clean version: “Our hypothesis in this study is that a high level of liver damage is indicative of a long-term chronic infection, which results also in flatter viral quasispecies in fitness terms. Using post-treatment quasispecies data from 19 patients who experienced failure with DAA regimens, and with fibrosis data available, we investigate the potential correlation between liver damage and quasispecies maturity and fitness. This is done characterizing the quasispecies in these patients through indicators of quasispecies maturity [4].”

The authors must improve the patients' presentations and the way they evaluate fibrosis. What stages are F4+? Are there more stages than cirrhosis (F4)? There are no data regarding the sex, age, or other lab results of the patients.

Answer:  We have added a table in supplementary materials with the known clinical data of the 19 patients used in the association study (Supplementary Table S1 in document “Main Supplementary Materials.pdf”) see line 180 in the clean version.

We have added Table S1 in the document named Main Supplementary Materials.pdf, and improved patients’ presentations in methods section 2.1, lines 104 to 108 in the clean version: “The full cohort from where the 19 patients with fibrosis scores have been taken, is composed of 75 patients, with failed HCV treatments. The 56 patients with no fibrosis data have been partitioned in three groups according to their rank with the RLEinf evenness indicator. Reg (18), Mid (19), and Top (19), with increasing maturity and flat-like fitness characteristics, according to their haplotype frequencies distribution.”

We have included a clear explanation on how transient elsastography was measured and which is the meaning of F4+. See lines 101 to 103 in the clean version:  “Transient elastography was measured using Fibroscan classifying the patients by the degree of fibrosis as F1, F2, F3 and F4+ (F4+ indicates advanced cirrhosis).”

Some data presented in the Results section are more appropriate to be in the Methodology section. 

Answer: We agree with reviewer and we have make the required rearrangements:

Previous section 3.1 has been moved to section 2.2.3 in Methods (lines 156 to 191  in the clean version).

Previous lines 162 to 164 in section 3.2 have been moved and improved to section 2.2.5 (lines 193 to 198 in the clean version)

Previous lines 165 to 172 in section 3.2 have been moved and improved in section 2.2.5 (lines 199 to 213 in the clean version)

Previous lines 178 to 182 in section 3.2, have been moved to section 2.2.6 (lines 223 to 229  in the clean version)

Previous lines 193 to 197 in section 3.3, have been moved to section 2.2.6 (lines 215 to 222  in the clean version)5

Table 2 may be changed. Instead of stg1 and stg 2, this may be split into 2 tables, and the comparison F3/F2 and F4/F3 results may be presented in each table.

Answer:  Done. A suggested by reviewer we have also divided the table into Table 4a comparing F3 vs F2 (line 268 in the clean version) and Table 4b to compare F4+ vs F3 (line 270 in the clean version)

All tables must include the definition of the abbreviated words.

Answer:  Done

Regarding the figures with AUC values, it would probably be significant to draw the level that shows the relevance of the AUC value for diagnostic purposes. 

Answer:  We have clarified this point in supplementary material providing guidelines to interpret the various effects size metrics used in the study. Please check Main Supplementary Materials.pdf

The authors should revise the text in lines 213 and 214 as there is no sentence there.

Answer:  Removed

It would be recommended not to use Fibroscan (as in line 291) but transient elastography.

Answer:  Done in line 374 in the clean version

The authors must include a paragraph describing their study's strengths and limitations. They may also discuss how the results will influence the management and care of these patients. 

Answer:  Thank you for the comment. We have added a paragraph in lines 333 to 352 discussing limitations and factors contributing to increase the variability in the results.

“The samples of the patients in this cohort correspond to samples sent to our laboratory with the purpose to get recommendations for a DAA combination treatment. Any previously failed treatment may have contributed to accelerate (RBV) or decelerate (DAA) the natural evolution in the quasispecies maturation process, and thus contribute to increase the variability in our data with respect to what could be observed with all naïve patients. One of the limitations of our study, is the lack of pre-treatment samples. However, it is likely that naive patients will form a cohort with lower variability, potentially leading to a more statistically significant p-values. In contrast, selecting patients treated with different therapies introduces greater variability, which may make it more challenging to detect correlations. Nevertheless, when correlations are found in such a diverse cohort, they may be more robust. While including pre-treatment samples would be valuable, we believe their absence will not significantly impact our reported results.

Moreover, mutagens contribute to accelerate the natural evolution through the increased replication error rate, DAAs contribute to decelerate the evolution through inhibition. We observed this behavior with cell culture experiments (26, and unpublished data). HCV genotypes/subtypes may follow the natural evolution at relatively different paces, although not big differences are expected, contributing also to increase the variability in our data. These possible extra sources of variability render our results more robust, although possibly at the expenses of increasing some p-values and reducing effect sizes.”

Round 2

Reviewer 2 Report

Comments and Suggestions for Authors

The authors have substantially improved their manuscript making it easier to read. They provide a manuscript examining the HCV quasispecies' structure in patients who failed DAA treatment.

The introduction is once again well writte, providing enough information to the reader in order to understand the manuscript. The materials and methods section is now clear, while the results section is also well written with good tables and figures. Lastly, the discussion section highlights the pros and cons of the manuscript and provides a solid conclusion part.

Comments on the Quality of English Language

Quality of English fine